# Thriving in a pandemic: Determinants of excellent wellbeing among New Zealanders during the 2020 COVID-19 lockdown; a cross-sectional survey

Ben Beaglehole[1]*, Jonathan Williman[2], Caroline Bell[1], James Stanley[3],
Matthew Jenkins[4], Philip Gendall[3], Janet Hoek[3], Charlene Rapsey[4], Susanna Every-
Palmer[5]

1 Department of Psychological Medicine, University of Otago, Christchurch, New Zealand, 2 Department of Population Health, University of Otago, Christchurch, New Zealand, 3 Department of Public Health, University of Otago, Wellington, New Zealand, 4 Department of Psychological Medicine, University of Otago, Dunedin, New Zealand, 5 Department of Psychological Medicine, University of Otago, Wellington, New Zealand

* ben.beaglehole@otago.ac.nz

**Data Availability Statement:** The anonymous data are available from the Dryad database licensed under a CCO 1.0 Universal Public Domain

## Abstract

### Objective

The COVID-19 pandemic and associated restrictions are associated with adverse psychological impacts but an assessment of positive wellbeing is required to understand the overall impacts of the pandemic.

### Methods

The NZ Lockdown Psychological Distress Survey is an on-line cross-sectional survey of 3487 New Zealanders undertaken during a strict lockdown for COVID-19. The lockdown extended from 25 March 2020 to 28 April 2020 and the survey was undertaken between 15 April 2020 and 27 April 2020. The survey measured excellent wellbeing categorised by a WHO-Five Well-being Index (WHO-5) score ≥22. The survey also contained demographic and pre-lockdown questions, subjective and objective lockdown experiences, and questions on alcohol use. The proportion of participants with excellent wellbeing is reported with multivariate analysis examining the relative importance of individual factors associated with excellent wellbeing.

### Results

Approximately 9% of the overall sample (303 participants) reported excellent wellbeing during the New Zealand lockdown. In the multivariable analysis, excellent wellbeing status was positively associated with increasing age (p<0.001), male gender (p = 0.044), Māori and Asian ethnicity (p = 0.008), and lower levels of education (certificate/diploma level qualification or less) (p<0.001). Excellent wellbeing was negatively associated with smoking (p =

Dedication License: https://doi.org/10.5061/dryad.
66t1g1k36.

**Funding:** The authors received no specific funding
for this work.

**Competing interests:** The authors have declared
that no competing interests exist.

0.001), poor physical (p<0.001) and mental health (p = 0.002), and previous trauma (p =
0.033).

## Conclusion

Nine percent of New Zealanders reported excellent wellbeing during severe COVID-19 pandemic restrictions. Demographic and broader health factors predicted excellent wellbeing status. An understanding of these factors may help to enhance wellbeing during any future lockdowns.

## Introduction

New Zealand identified its first COVID-19 case on 28 February 2020. Case numbers increased during March 2020 and New Zealand (NZ) entered alert level 4 (a stringent lockdown) on 11.59pm, 25 March 2020. All schools and non-essential businesses closed and, except for essential workers, citizens were required to stay at home (except when undertaking essential shopping, health care, and exercise). The lockdown ended on 28 April. The restrictions were considered some of the strictest imposed globally [1] and successfully eliminated COVID-19 from NZ for a period of time.

The lockdown restrictions reduced family and social contact, limited recreation opportunities, caused job losses and financial insecurity; and restricted attendance at educational, religious and social sites. Recent studies have identified that the COVID-19 pandemic and its associated restrictions are associated with negative psychological effects, including increased psychological distress, increased suicidal ideation, and increased risk of mental disorder in those assessed (2–4). The literature also suggests that the impacts of the COVID-19 pandemic are not uniformly distributed. Sub-groups facing particular issues include parents for whom parenting exhaustion is a concern [2], LGBT individuals [3], and students [4]. These findings suggest that sub-group analyses are required to attain a detailed as opposed to population-level understanding of the psycho-social impacts of the pandemic.

Trauma and disasters can also lead to post-traumatic growth and thriving where individuals and groups do well despite adverse experiences [5, 6] including previous epidemics [7]. This suggests that a full understanding of the COVID-19 pandemic's effects requires assessment of positive as well as negative consequences. Although some studies have reported post-traumatic growth during the COVID-19 pandemic [8, 9], we are unaware of studies examining for the possibility that some people will experience excellent wellbeing despite restrictions. We therefore measured New Zealanders' wellbeing during the COVID-19 lockdown to identify factors associated with excellent wellbeing status during this period.

## Methods

The NZ Lockdown Psychological Distress Survey is an online cross-sectional survey of 3487 New Zealanders undertaken between 15 April 2020 and 27 April 2020 during level 4 lockdown. The study was granted ethical approval by the University of Otago Human Ethics Committee (approval code F20/003) and was reviewed by the Ngāi Tahu Research Committee. All participants were asked to read a participant information sheet and gave written informed consent before they could proceed with the survey.

We aimed to recruit a sample that represented the New Zealand adult population. Recruitment occurred via two pathways. Firstly, we used a commercial survey platform (Dynata) which invited participants from their commercial survey panel and applied target participation quotas by age, sex, and ethnicity [10]. Secondly, we invited participation from New Zealanders who had previously been randomly selected by the NZ Ministries of Health and Justice to participate in large-scale national data surveys [11, 12] and who had consented to further contact. Population-level outcomes for the Dynata sample have been reported previously [10]. The survey was completed on-line. Analytical methods required to accommodate these sampling steps are detailed below.

## Survey questionnaire

The WHO-Five Well-being Index (WHO-5) was used to measure wellbeing [13]. The WHO-5 is a 5-item scale; each question evaluates a different measure of positive wellbeing using a six-item Likert scale ranging from 0 (not present) to 5 (constantly present). Although low scores on the WHO-5 are used as a screen for depression, the WHO-5 scale is regarded as a measure of mental wellbeing rather than just the absence of depressive symptoms [14]. Various cut points are reported to define high and excellent wellbeing [15, 16]. We used the cut points of Yallop et al. [17] for poor (score<13), good (score 13–17), very good (score 18–21) and excellent (score 22–25) wellbeing. Our focus is on those who thrived despite the lockdown experience. We therefore compare those with excellent wellbeing compared to those with lower scores (WHO-5 scores <22).

The survey questions are available as a supplementary file (S1 File). The survey assessed demographic and pre-lockdown socio-economic factors, objective and subjective lockdown experiences, substance use, psychological distress, and wellbeing.

Demographic and pre-lockdown factors included age, gender, ethnicity, socio-economic status (education and household income), employment, smoking and alcohol usage, general and mental health, and prior trauma.

We assessed objective lockdown experiences using questions on living circumstances during lockdown, essential worker status, contacts with others, work load during lockdown, and COVID-19 exposure status. These questions explored the participant's bubble; defined as the people sharing the household with the participant during the lockdown. We examined respondents' contacts with others outside their bubble via written, electronic, and face to face media, and summarised contact frequency as high, medium, and low contact. We also asked if contact with others outside the bubble had increased, decreased, or stayed the same since the lockdown began.

We explored subjective lockdown experiences through questions on satisfaction with lockdown home environment, personal relationships during lockdown, stressors, and concerns about risk of infection. Respondents were also asked if they had experienced 'silver linings' personally or for society as a result of the lockdown experience.

Alcohol use and smoking were assessed by questions on the amount consumed before and during the lockdown.

Further contextual information on wellbeing among adults in NZ is drawn from StatsNZ's General Social Survey (NZGSS) [18], which collects data on the well-being of New Zealanders aged 15 years and over. Estimates of excellent wellbeing status (score 22–25) were requested from StatsNZ to provide a national baseline of wellbeing data prior to the pandemic. As excellent wellbeing data from the NZGSS is not in the public domain, we report these findings in our results.

## Statistical analysis

This paper presents a secondary analysis of data collected during national surveys. The primary purpose of this analysis is to estimate associations between variables and draw inferences that may be applied to wide range of populations. We therefore combined the Dynata and Ministry datasets and used unweighted data to increase the sample size to improve statistical precision of estimates. Prior to combining datasets, we ran the analyses separately to ensure consistency of effects.

The exception to this approach occurred when we compared weighted prevalence estimates of excellent wellbeing in the Dynata and Ministry datasets to pre-COVID pandemic results from the New Zealand General Social Survey [18]. Details of the weighting strategy used for the Dynata and Ministry datasets is described in our parent paper [10]. For this comparison, national prevalence estimates are presented separately for the Dynata and Health and Justice survey panel datasets due to the differing sampling strategies and sampling weights applied to these surveys. Statistical tests were not undertaken for this comparison due to the differences in sampling strategies between the NZGSS and the study groups included in this paper.

We grouped potential explanatory factors into demographic and pre-lockdown factors, objective lockdown factors, and subjective lockdown experiences. The proportion of respondents reporting excellent wellbeing was calculated for each explanatory factor. We assessed differences across levels using chi-squared or Fisher's exact tests. Unadjusted odds ratios with 95% confidence intervals were calculated for selected pairwise differences versus a nominated reference category. We explored the relative importance of individual factors by creating a series of four nested multivariable logistic regression models, entering variables in four blocks; demographics (age, sex, ethnicity); qualifications and employment; pre-existing risk factors (smoking status, prior mental health diagnosis, self-rated health, past history of trauma); and household composition. We did not incorporate subjective and objective lockdown experiences into this model due to the risk of reverse causality. There were 19 participants did not complete the WHO-5 and are excluded from analysis. A further 106 participants had missing data for some of the variables. When this occurred, participants were excluded from the multivariable analysis but are still included in descriptive and univariable analyses. Analysis was performed using the R 4.0.3 programming language and environment [19].

## Results

### Demographic and socioeconomic factors

A total of 3,487 participants completed the surveys (2010 from the Dynata survey and 1477 from the Ministries of Health and Justice dataset). Nineteen participants did not provide full WHO-5 data and are excluded from the analysis. 32.8% (n = 1139) of the sample were in the poor wellbeing group (WHO-5<13), 67.2% (n = 2329) were rated good wellbeing or better (WHO-5≥13) and, among the latter, 303 participants (8.7%) were in the excellent wellbeing group. The overall mean WHO-5 score was 14.7 (SD = 5.71).

Females were significantly less likely than males to report excellent wellbeing (OR = 0.73, CI 0.58–0.93, p = 0.010). Older people, particularly those over 65, were more likely than young people to report excellent wellbeing (OR for 65+ years compared to 15–24 years = 2.64, CI 1.63–4.28, p<0.001). People with higher qualification levels (Bachelor's degree or greater) were less likely to report excellent wellbeing compared people with lower levels (OR = 0.49, CI 0.33–0.73, p<0.001). The retired were more likely to report excellent wellbeing than those who were working (OR = 2.01, CI 1.54–2.62, p<0.001)) but the unemployed were less likely to report excellent wellbeing than workers (OR = 0.64, CI 0.43–0.96, p = 0.031). Excellent wellbeing status was comparable between essential and non-essential workers (OR = 1.16, CI 0.84–

1.60, p = 0.402). Self-rated good or better physical health status was strongly associated with excellent wellbeing (OR = 4.84, CI 2.86–8.19, p<0.001). Pre-COVID-19 mental illness (OR = 0.29, CI 0.19–0.47, p<0.001), physical illness (OR = 0.52, CI 0.30–0.91, p = 0.020), and prior trauma (OR = 0.75, CI 0.57–0.99, 0 = 0.042) were all negatively associated with excellent wellbeing. There were no significant effects by ethnicity (Fishers exact test p = 0.510) or income (chi-squared p = 0.780) on excellent wellbeing status.

## Objective lockdown experiences

The composition of the bubble was not significantly associated with excellent wellbeing (chi-squared p = 0.057). Connections with others were significantly associated with excellent wellbeing with highest rates of excellent wellbeing being reported by people with a high frequency of connection with others (chi-squared p = 0.012). Reducing the frequency of contact with others was associated with lower rates of excellent wellbeing relative to those whose frequency of contact with others remained unchanged (OR = 0.39, CI 0.28–0.54, p<0.001). There were no significant effects for change in work load (Fishers exact test p = 0.364) or loss of job during COVID-19 lockdown (Fishers exact test p = 0.101) or between COVID-19 exposure or infection status on the likelihood of excellent wellbeing (Fishers exact test p = 0.196).

## Subjective lockdown experiences

Those who were extremely satisfied with their 'bubbles' were more likely to report excellent wellbeing than those who were not satisfied (OR = 3.17, CI 2.04–4.93, p<0.001). Similarly, getting along very well with others in their 'bubbles' was associated with excellent wellbeing compared to those who were not getting on well with others (OR = 6.41, CI 3.62–11.35, p<0.001). Conversely, greater loneliness was associated with a reduced proportion reporting excellent well-being, relative to people who did not feel lonely (OR = 0.11, CI 0.07–0.18, p<0.001). Looking at information on COVID-19 for more than two hours/day was not significantly associated with excellent wellbeing status (OR = 0.73, CI 0.53–1.11, p = 0.165). However, stress about personal health (OR = 0.43, CI 0.28–0.64, p<0.001), the health of loved ones (OR = 0.34, CI 0.25–0.47, p<0.001), finances (OR = 0.29, CI 0.20–0.41, p<0.001), employment security (OR = 0.56, CI 0.39–0.79, p<0.001), and the wider consequences of COVID-19 (OR = 0.42, CI 0.32–0.54, p<0.001) were all associated with reduced likelihood of reporting excellent wellbeing. Reporting 'silver linings' during the COVID-19 lockdown, either personally (OR = 1.02, CI 0.80–1.29, p = 0.904) or societal-level silver linings (OR = 0.98, CI 0.77–1.24, p = 0.854), was not significantly associated with excellent wellbeing status.

## Substance use

Smokers (current and former) were less likely to report excellent wellbeing than those who never smoked (OR for current compared to never smoked = 0.48, CI 0.32–0.72, p<0.001). There were no significant relationships between pre-lockdown drinking levels and excellent wellbeing status although hazardous drinking during the lockdown (OR = 0.63, CI 0.42–0.96, p = 0.029) and increasing (OR = 0.43, CI 0.31–0.60, p<0.001) or decreasing alcohol (OR 0.70 CI 0.50–0.97, p = 0.031) intake during the lockdown were associated with a reduced likelihood of excellent wellbeing.

## Multivariable analysis/logistic regression

Multivariable modelling is reported in Table 1. Following the multivariate analysis; excellent wellbeing was independently associated with older age, male gender, Māori and Asian

**Table 1. Multivariable modelling of independent variables and excellent wellbeing.**

| | | Respondents | Excellent wellbeing | Adjusted | |
|---|---|---|---|---|---|
| Characteristic | Level | N | % (n) | Odds ratio (95% CI) | p |
| Age (years) | 15–24 | 322 | 6.5 (21) | 1.00 (Reference) | <0.001 |
| | 25–34 | 582 | 4.6 (27) | 0.79 (0.43, 1.47) | |
| | 35–44 | 588 | 4.8 (28) | 0.93 (0.52, 1.72) | |
| | 45–54 | 611 | 9.2 (56) | 1.83 (1.10, 3.19) | |
| | 55–64 | 593 | 8.6 (51) | 1.73 (1.02, 3.07) | |
| | 65+ | 772 | 15.5 (120) | 3.02 (1.71, 5.53) | |
| Gender | Male | 1476 | 10.2 (150) | 1.00 (Reference) | 0.044 |
| | Female | 1970 | 7.7 (151) | 0.80 (0.65, 0.99) | |
| | Unknown | 22 | - | - | |
| Ethnicity | European/other | 2381 | 8.5 (202) | 1.00 (Reference) | 0.008 |
| | Maori | 620 | 8.4 (52) | 1.38 (1.00, 1.85) | |
| | Pacific | 148 | 10.8 (16) | 1.43 (0.79, 2.32) | |
| | Asian | 319 | 10.3 (33) | 1.86 (1.25, 2.68) | |
| Qualification | None | 375 | 11.5 (43) | 1.00 (Reference) | <0.001 |
| | High school | 960 | 10.3 (99) | 0.84 (0.60, 1.20) | |
| | Certificate or Diploma | 871 | 9.8 (85) | 0.86 (0.61, 1.24) | |
| | Bachelors or higher | 1262 | 6 (76) | 0.49 (0.34, 0.73) | |
| Employment | Employed | 2230 | 7.9 (176) | 1.00 (Reference) | 0.703 |
| | Unemployed | 576 | 5.2 (30) | 0.85 (0.56, 1.24) | |
| | Retired | 661 | 14.7 (97) | 0.98 (0.71, 1.38) | |
| | Unknown | 1 | - | - | |
| Smoking | Never smoked | 1873 | 10.1 (190) | 1.00 (Reference) | 0.001 |
| | Past | 1047 | 8.1 (85) | 0.70 (0.54, 0.89) | |
| | Current | 545 | 5.1 (28) | 0.57 (0.37, 0.84) | |
| | Unknown | 3 | - | - | |
| Self-reported Health | Poor/fair | 652 | 2.3 (15) | 1.00 (Reference) | <0.001 |
| | Good/excellent | 2816 | 10.2 (288) | 3.83 (2.33, 6.87) | |
| Prior mental health | No | 2763 | 10 (277) | 1.00 (Reference) | 0.002 |
| | Yes | 629 | 3.2 (20) | 0.53 (0.33, 0.81) | |
| | Unknown | 76 | - | - | |
| Physical disability | No | 3186 | 9.1 (289) | 1.00 (Reference) | 0.335 |
| | Yes | 282 | 5 (14) | 0.78 (0.43, 1.27) | |
| Prior trauma | No | 2432 | 9.4 (228) | 1.00 (Reference) | 0.033 |
| | Yes | 1036 | 7.2 (75) | 0.77 (0.59, 0.98) | |
| Household compostion | Solo | 527 | 10.2 (54) | 1.00 (Reference) | 0.636 |
| | Two adults | 1103 | 10.1 (111) | 0.91 (0.68, 1.23) | |
| | Multiple adults | 685 | 7.2 (49) | 0.85 (0.58, 1.25) | |
| | With children | 1149 | 7.7 (89) | 1.04 (0.73, 1.49) | |
| | Unknown | 4 | - | - | |

ethnicity, and certificate/diploma level qualification or less. Excellent wellbeing was also negatively associated with smoking, poor physical and mental health, and previous trauma.

## 2018 New Zealand General Social Survey (NZGSS) pre-COVID-19 comparison

Table 2 reports excellent wellbeing separately for the Dynata panel and the Health and Justice survey panel compared to the NZGSS. Rates of excellent wellbeing group were 8.7% (Dynata

**Table 2. Excellent wellbeing status: Comparison with WHO-5 data from the 2018 NZGSS.**

| Characteristic | Level | Dynata dataset % | CI | Health and Justice surveys % | CI | NZGSS % | CI |
|---|---|---|---|---|---|---|---|
| Total | | 8.7 | (7.5, 10.1) | 7.8 | (6.4, 9.5) | 7.0 | (5.6, 8.4) |
| Gender | Male | 9.5 | (7.7, 11.7) | 8.1 | (6.1, 10.7) | 8.1 | (5.6, 10.6) |
| | Female | 8.0 | (6.3, 9.9) | 7.5 | (5.7, 9.8) | 5.9 | (4.1, 7.7) |
| Age | 15–24 | 6.6 | (4.1, 10.4) | 2.9 | (0.7, 11.9) | 8.4 | (4.3, 12.5) |
| | 25–34 | 4.9 | (3.1, 7.7) | 4.2 | (1.9, 8.9) | 6.7 | (2.2, 11.2) |
| | 35–44 | 5.1 | (3.2, 8.1) | 5.4 | (2.9, 10.1) | 6.3 | (2.2, 10.4) |
| | 45–54 | 9.8 | (6.9, 13.7) | 8.5 | (5.3, 13.4) | 4.9 | (1.2, 8.6) |
| | 55–64 | 8.6 | (5.8, 12.6) | 8.7 | (5.6, 13.1) | 4.1 | (1.9, 6.3) |
| | 65+ | 16.1 | (12.3, 20.9) | 15.4 | (13.4, 22.0) | 10.5 | (5.6, 15.4) |
| Ethnicity | European/Other | 7.8 | (6.4, 9.5) | 8.1 | (6.6, 9.8) | 5.9 | (4.5, 7.3) |
| | Maori | 8.7 | (6.2, 12.1) | 5.6 | (3.3, 9.5) | 7.0 | (2.9, 11.1) |
| | Pacific | 8.8 | (4.7, 15.8) | 10.9 | (3.7, 28.4) | 8.5 | (3.2, 13.8) |
| | Asian | 12.6 | (8.5, 18.3) | 7.6 | (3.3, 16.6) | 10.6 | (4.9, 16.3) |

% are weighted according to the NZ population

sample) and 7.8% (Health and Justice panel) compared to 7.0% percent of the NZGSS sample. This pattern was largely repeated when data was broken down by gender and ethnicity although Māori participants in the Health and Justice survey panel reported lower rates of excellent wellbeing than the Dynata and NZGSS datasets. Rates of excellent wellbeing for those aged under 45 years were consistently lower in the post-COVID-19 datasets compared to the NZGSS.

## Discussion

Nine percent of the survey population reported excellent wellbeing during the lockdown. There was substantial pattering across different population groups. Male gender, older age, Māori and Asian ethnicity, lower levels of education, and being a non-smoker were associated with excellent wellbeing status. Males typically score higher on the WHO-5 index than females [14, 20]; our finding is consistent with this and epidemiological studies that report higher rates of mood and anxiety disorders for females (for example [21, 22]). Rates of excellent wellbeing varied across ethnic groups. Māori and Asian participants were more likely to be in the excellent wellbeing group than the NZ European/Other group. This may relate to the importance of family connections for these groups compared to non-Europeans [23]. Further studies are required to see if this finding is repeated or relates to sampling methods specific to this study.

We expected that the elderly might be more negatively affected by COVID-19 restrictions. However, those aged 65+ were 2.64 times more likely to report excellent wellbeing than those aged 15–24. We speculate that the social isolation was particularly difficult for the younger age group and was not compensated for by non-face-to-face social connections. It is also possible that the elderly were relatively protected from the economic consequences of the pandemic. The comparison with pre-COVID-19 NZGSS data suggests that the 65+ age group were also more likely to report excellent wellbeing prior to the COVID-19 pandemic. A study of older individuals in Germany also reported that mental wellbeing was largely unaffected by a COVID-19 lockdown [24]. These findings suggest that younger age groups rather than the elderly require specific attention when planning interventions to mitigate adverse impacts of future lockdowns.

The finding that excellent wellbeing was related to lower levels of education was unexpected. We are aware of studies reporting that poor educational achievement in school is linked to subsequent risk of mental health problems [25] and assumed that this link would be reflected in lower wellbeing among those with less education. We were unable to identify literature reporting on levels of education and wellbeing to see if our findings are reproduced elsewhere.

The relationships between frequency and quality of contact with others, satisfaction with bubbles, and loneliness all point to the importance of quality social connections. These findings are a reminder of the importance of links between social capital (including formal and informal social interactions) and wellbeing [26]. Although our data is cross-sectional in nature (reducing our ability to infer causal links between these factors and wellbeing), bolstering the ability for formal and informal social connections to occur during lockdowns could be considered by governments and support agencies to mitigate adverse effects caused by social restrictions and isolation.

Prior physical and mental health issues, and prior trauma are static risk factors that are difficult to modify. However, the range of stressors (health, finances, employment, and COVID-19) associated with reduced likelihood of excellent wellbeing provides direction for public health initiatives in both preventative and responsive frameworks. Clear communication from governments and appropriate safety nets for those affected by loss of employment may have a role in mitigating adverse effects and improving wellbeing. Similarly, health messaging around stopping smoking and drinking within recommended limits appears relevant during lockdowns as well as other times.

## Limitations

We merged data collected by Dynata with data from the NZ Ministries of Health and Justice for the analysis. This approach increased our sample size and allowed us to make more precise estimates of these associations for the larger sample. This strategy means any between dataset differences may not be highlighted although the analyses were run separately prior to merging to minimise the risk of important differences being missed.

Our data are cross-sectional and the associations we report do not allow strong causal inferences to be made. Although our sampling strategy used quotas to achieve a demographically representative sample, our respondents were computer literate, and were both available to participate and consented to complete the survey. Despite being demographically representative, the study population may not therefore be representative of the national population.

The COVID-19 infection rate in NZ was low over the study period. We therefore assessed the impacts of the NZ lockdown and fear of infection as opposed to assessing wellbeing in the presence of high rates of COVID-19 community transmission. We completed data collection in the early stages of the pandemic. We therefore provide a snapshot of wellbeing at that point of time and recognise the importance of repeated measures in order to track progress over time.

## Conclusion

Nine percent of the survey population reported excellent wellbeing during the lockdown compared to 7% of those surveyed prior to the lockdown in the NZGSS. This suggests that focussing on the negative consequences of lockdown restrictions does not provide a balanced understanding of the psychosocial impacts of pandemic restrictions. Some of the factors associated with excellent wellbeing were static and not amenable to change. Other factors highlighted the importance of stress, substance use, health, and relationships in determining wellbeing and provide a path for governments and individuals to enhance wellbeing.

## Supporting information

**S1 File. The NZ Lockdown Psychological Distress Survey questionnaire.**
(PDF)

**S2 File. Inclusivity in global research questionnaire.**
(DOCX)

## Acknowledgments

We would like to thank Dynata, the NZ Ministry of Health, Ministry of Justice, and the Department of Statistics for their generous support of this research. We also thank Anaru Waa, Emma Sutich, Marcellus Paki, Fiona Mathieson, Giles Newton-Howes, and Elliot Bell for expert advice on survey content and design.

## Author Contributions

**Conceptualization:** Ben Beaglehole, Matthew Jenkins, Philip Gendall, Susanna Every-Palmer.

**Data curation:** Jonathan Williman, James Stanley.

**Formal analysis:** Jonathan Williman, James Stanley.

**Methodology:** Ben Beaglehole, Jonathan Williman, Caroline Bell, James Stanley, Matthew Jenkins, Philip Gendall, Janet Hoek, Charlene Rapsey, Susanna Every-Palmer.

**Writing – original draft:** Ben Beaglehole.

**Writing – review & editing:** Ben Beaglehole, Jonathan Williman, Caroline Bell, James Stanley, Matthew Jenkins, Philip Gendall, Janet Hoek, Charlene Rapsey, Susanna Every-Palmer.

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
