## [Decision Letter · Decision Letter 0]

6 Dec 2021

PONE-D-21-34860Thriving in a pandemic: determinants of excellent wellbeing among New Zealanders during the 2020 COVID-19 lockdownPLOS ONE

Dear Dr. Ben Beaglehole,

Thank you for submitting your manuscript to PLOS ONE. After careful consideration, we feel that it has merit but does not fully meet PLOS ONE’s publication criteria as it currently stands. Therefore, we invite you to submit a revised version of the manuscript that addresses the points raised during the review process.

We look forward to receiving your revised manuscript.

Kind regards,

Rajnish Joshi

Academic Editor

PLOS ONE

Journal Requirements:

(We would like to thank Dynata for their generous support of this research. Thank you to the NZ Ministry of Health, Ministry of Justice, Department of Statistics, and to Anaru Waa, Emma Sutich, Marcellus Paki, Fiona Mathieson, Giles Newton-Howes, and Elliot Bell for expert advice on survey content and design.)

(The authors received no specific funding for this work.)

7. Please upload a copy of Supporting Information S1 which you refer to in your text on page 4.

Reviewers' comments:

Reviewer's Responses to Questions

**Comments to the Author**

1. Is the manuscript technically sound, and do the data support the conclusions?

Reviewer #1: Yes

Reviewer #2: Partly

2. Has the statistical analysis been performed appropriately and rigorously? 

Reviewer #1: Yes

Reviewer #2: Yes

3. Have the authors made all data underlying the findings in their manuscript fully available?

Reviewer #1: No

Reviewer #2: Yes

4. Is the manuscript presented in an intelligible fashion and written in standard English?

Reviewer #1: Yes

Reviewer #2: Yes

5. Review Comments to the Author

Reviewer #1: The paper entitled: “Thriving in a pandemic: determinants of excellent wellbeing among New Zealanders during the 2020 COVID-19 lockdown” explores the results of a survey on the wellbeing of New Zealanders during the lockdown. The main results are 9that % of the sample reported excellent wellbeing during the New Zealand lockdown, and was associated with older age, male gender, Māori and Asian ethnicity, and lower levels of education. On the other hand, it is negatively associated with smoking, poor physical and mental health, and previous trauma. The paper concludes that a substantial minority of New Zealanders reported excellent wellbeing during severe COVID-19 pandemic restrictions.

I think the paper makes a nice job in adding a specific contribution to the related literature, giving the scholars’ community some new information about the well-being of New Zealanders during the 2020 lockdown. The nested model and the summary statistic presented are up to the point and make a good job in giving to the scholars an idea of the data. The results are interesting. I would nonetheless like to see some amendments to improve the paper.

First, the authors make use of unweighted data in the merge (line 117). I would like to see more explanation about this choice and caveat about the possible consequences of this in terms of representativeness of the sample and external validity of the results.

Second, I see room for improvement in the literature review. The authors cite in the article relevant articles in the introduction, this can be expanded. First, some articles on the effects of COVID-19 lockdown on mental health seem appropriate, such as Evans et al., 2021, Sharma and Subramanyam, 2021), and Rohr et al., 2020. Moreover, articles discussing the possible impact of other cultural and socio-economic characteristics on lockdown compliance seem to be relevant, such as Alfano, 2021 for work ethics, or Marchetti et al. (2020) for being parents. This would help better frame the article in the literature and enhance its collocation in the literature.

I wish best luck to the authors with this interesting piece of research!

References

Alfano, V. (2021). Work ethics, stay-at-home measures and COVID-19 diffusion. Eur J Health Econ.

Evans, S., Erkan Alkan, Jazmin K. Bhangoo, Harriet Tenenbaum, Terry Ng-Knight, Effects of the COVID-19 lockdown on mental health, wellbeing, sleep, and alcohol use in a UK student sample, Psychiatry Research,

Volume 298, 2021, 113819.

Marchetti, D., Lilybeth Fontanesi, Mazza, C., Di Giandomenico, S., Roma, P., Verrocchio, M.C. (2020). Parenting-Related Exhaustion During the Italian COVID-19 Lockdown, Journal of Pediatric Psychology, Volume 45, Issue 10, Pages 1114–1123.

Röhr, S., Reininghaus, U. & Riedel-Heller, S.G. Mental wellbeing in the German old age population largely unaltered during COVID-19 lockdown: results of a representative survey. BMC Geriatr 20, 489.

Sharma, A. J., Subramanyam, M. A. (2021). A cross-sectional study of psychological wellbeing of Indian adults during the Covid-19 lockdown: Different strokes for different folks.

Reviewer #2: The topic of this study is interesting. I have following comments for the authors.

TITLE:

Please add the study design (i.e. A cross sectional survey) at the end of the title according to the The EQUATOR Network guidelines for reporting survey. Please refer to the 'A Consensus-Based Checklist for Reporting of Survey Studies (CROSS)' available https://www.equator-network.org/reporting-guidelines/a-consensus-based-checklist-for-reporting-of-survey-studies-cross/

ABSTRACT:

Methods: please report the study design, total sample size, sampling methods and how the survey was administered. Also, please could you report the dates/duration (months and year) of lockdown that was studied in this study.

Results: Please report the number of respondents that accounted for 9% of the sample.

Results: Could you please report the age range that refers to 'older age' as well as what was the cut off education level that is reported as lower levels of education.

Results. Please report the evidence i.e. statistics showing significant associations between reported variable.

Conclusion: The authors conclude that substantial minority reported excellent wellbeing. Is 9% a substantial figure? Please revise your conclusion remarks.

INTRODUCTION

Citations: Please merge the citations in the text because they are mostly included as standalone numbers such as (2) (3) (4), which should be (2-4). Please do this in the whole manuscript.

METHODS

Sample: Please double check the number of participants because 2020 (Dynata)+1477 (MoHJ) = 3497. But you have reported total 3487 participants. In addition, you have reported that 120 cases with missing data were excluded which means 3497-120= 3377. But your final figures do not match this calculation.

Results:

Total participants: Please see comment about the number of participants as mentioned in the methods section above and revise your results accordingly.

Please check you number of total respondents and numbers given in Table 1.In there were missing values for example in the gender etc then please report these in Table 1. For example gender (Male and female = 1476+1970=3446. This number does not match with the total respondents reported at the beginning of the results section. Please check for this for all variables and categories.

Please report the number / count of participants along with the % figures.

Please report Standard deviation values along with the mean values and if data was no normally distributed then report the median number too.

Please sign post about the tables in the text.

Please report p -value along with the Odds Ratios.

Please clarify/report whether the lower education level included people who had education less than a Bachelor's degree or something else. Please be specific.

Please report statistics or refer to the relevant table in the sentences where you report associations. for example, there is no such information in this sentence: "Connections with others were significantly associated with excellent wellbeing....".,

Table 2: Not sure why data from the 2018 NZGSS has been included in the manuscript> It could be referred in the discussion only.

6. PLOS authors have the option to publish the peer review history of their article (what does this mean?). If published, this will include your full peer review and any attached files.

Reviewer #1: No

Reviewer #2: **Yes: **Syed Ghulam Sarwar Shah

---

## [Author Response · Author response to Decision Letter 0]

19 Dec 2021

13.12.21

Response to reviewers

Dear Dr Joshi

Thank-you for the opportunity to revise Thriving in a pandemic: determinants of excellent wellbeing among New Zealanders during the 2020 COVID-19 lockdown; a cross-sectional survey. 

Response to editorial questions

1. We have reviewed the style requirements and believe these are met.

2. We have included the Inclusivity of global research questionnaire as a Supporting Information document. 

3. We have amended our acknowledgements statement to make the relationship with Dynata more clear. We were grateful for their support of the project but they did not provide any funding. The study was completed without funding so we do not need to update the Funding Statement.

4. Data availability statement; we will place the data in a repository if accepted for publication but have not done so at this stage.

5. Captions for Supporting Information files are now included.

6. Supporting Information S1 file has now been uploaded.

We thank Reviewers 1 and 2 for their constructive comments. We believe their feedback has improved the paper. Our responses to their comments follows.

Reviewer 1

1. We have expanded on our original explanation relating to the merging of the datasets in the methods. We have also provided additional comment in the limitations section of the discussion to provide context to the decision to merge data.

2. We have expanded the introduction as suggested to provide better context to our research. We have also referenced the Rohr paper in the discussion. The tracked changes version highlights the areas of change.

Reviewer 2

1. Title: We have added ‘cross-sectional survey’ to the study title as per the CROSS checklist.

2. Abstract: Methods; we have added details of the design, sample size, and survey administration to the methods. We have also provided the lockdown dates.

3. Abstract: Results; we have added the number of respondents (303). We have also specified increasing age (this finding was part of the multivariable analysis) and detailed the level of education (certificate/diploma level or under). We have also clarified that the findings relate to multivariable analysis and provided p values.

4. Abstract: Conclusion; we have removed ‘substantial minority’ and replaced this with the more accurate ‘Nine percent’.

5. We have merged the citations as suggested.

6. Methods: Sample; the sample size was 3487 as reported but we made an error in reporting the Dynata sample (this should be 2010). We have corrected this in the results. Nineteen participants did not provide full WHO-5 data and were excluded from the analysis. A total of 106 participants had missing data for some of the other variables. These participants are included in the descriptive and univariable analyses but are excluded from the multivariable analysis. We have clarified this in the methods section and results section.

7. Results: Total participants; see above for details.

8. Results: Missing values now included in Table 1.

9. Results: We have amended the reporting of % figures to include the number of participants.

10. We have now included the SD for the mean WHO-5 score

11. Results: Line 238 signposts Table 1 in the text

12. Results: We have added the p-values for the reported statistics.

13. Results: We have clarified that the lower education level is certificate/diploma level qualification or less.

14. Results: We have now provided the relevant statistics when associations are reported.

15. Results: Table 2. We have provided more explanation in the methods. The data we report from the NZGS is not in the public domain. Further analysis of the NZGSS data was required to report excellent wellbeing. We therefore believe that this data belongs in the results section.

---

## [Decision Letter · Decision Letter 1]

5 Jan 2022

Thriving in a pandemic: determinants of excellent wellbeing among New Zealanders during the 2020 COVID-19 lockdown; a cross-sectional survey

PONE-D-21-34860R1

Dear Dr. Ben Beaglehole,

We’re pleased to inform you that your manuscript has been judged scientifically suitable for publication and will be formally accepted for publication once it meets all outstanding technical requirements.

Kind regards,

Rajnish Joshi

Academic Editor

PLOS ONE

Additional Editor Comments (optional):

Reviewers' comments:

Reviewer's Responses to Questions

**Comments to the Author**

1. If the authors have adequately addressed your comments raised in a previous round of review and you feel that this manuscript is now acceptable for publication, you may indicate that here to bypass the “Comments to the Author” section, enter your conflict of interest statement in the “Confidential to Editor” section, and submit your "Accept" recommendation.

Reviewer #1: All comments have been addressed

Reviewer #2: All comments have been addressed

2. Is the manuscript technically sound, and do the data support the conclusions?

Reviewer #1: Yes

Reviewer #2: Yes

3. Has the statistical analysis been performed appropriately and rigorously? 

Reviewer #1: Yes

Reviewer #2: Yes

4. Have the authors made all data underlying the findings in their manuscript fully available?

Reviewer #1: Yes

Reviewer #2: No

5. Is the manuscript presented in an intelligible fashion and written in standard English?

Reviewer #1: Yes

Reviewer #2: No

6. Review Comments to the Author

Reviewer #1: My comments have been addressed, and the paper is much better. I wish to the authors best luck with this interesting piece of research.

Reviewer #2: Thanks for addressing all issues raised in my earlier review report. The manuscript is clear and improved now. Data could be made available or any relevant statement should be included in the manuscript.

7. PLOS authors have the option to publish the peer review history of their article (what does this mean?). If published, this will include your full peer review and any attached files.

Reviewer #1: No

Reviewer #2: **Yes: **Syed Ghulam Sarwar Shah

---

## [Editor Report · Acceptance letter]

23 Feb 2022

PONE-D-21-34860R1 

Thriving in a pandemic: determinants of excellent wellbeing among New Zealanders during the 2020 COVID-19 lockdown; a cross-sectional survey 

Dear Dr. Beaglehole:

I'm pleased to inform you that your manuscript has been deemed suitable for publication in PLOS ONE. Congratulations! Your manuscript is now with our production department. 

Kind regards, 

on behalf of

Dr. Rajnish Joshi 

Academic Editor

PLOS ONE